# Information-Theoretic Models for Physical Observables

**DOI:** 10.3390/e25101448

**Published:** 2023-10-14

**Authors:** D. Bernal-Casas, J. M. Oller

**Affiliations:** Department of Genetics, Microbiology and Statistics, Faculty of Biology, Universitat de Barcelona, 08028 Barcelona, Spain

**Keywords:** information geometry, Fisher’s information, Riemannian manifolds, Schrödinger’s equation, principle of minimum Fisher’s information, quantum harmonic oscillator, Bayes’ theorem

## Abstract

This work addresses J.A. Wheeler’s critical idea that all things physical are information-theoretic in origin. In this paper, we introduce a novel mathematical framework based on information geometry, using the Fisher information metric as a particular Riemannian metric, defined in the parameter space of a smooth statistical manifold of normal probability distributions. Following this approach, we study the stationary states with the time-independent Schrödinger’s equation to discover that the information could be represented and distributed over a set of quantum harmonic oscillators, one for each independent source of data, whose coordinate for each oscillator is a parameter of the smooth statistical manifold to estimate. We observe that the estimator’s variance equals the energy levels of the quantum harmonic oscillator, proving that the estimator’s variance is definitively quantized, being the minimum variance at the minimum energy level of the oscillator. Interestingly, we demonstrate that quantum harmonic oscillators reach the Cramér–Rao lower bound on the estimator’s variance at the lowest energy level. In parallel, we find that the global probability density function of the collective mode of a set of quantum harmonic oscillators at the lowest energy level equals the posterior probability distribution calculated using Bayes’ theorem from the sources of information for all data values, taking as a prior the Riemannian volume of the informative metric. Interestingly, the opposite is also true, as the prior is constant. Altogether, these results suggest that we can break the sources of information into little elements: quantum harmonic oscillators, with the square modulus of the collective mode at the lowest energy representing the most likely reality, supporting A. Zeilinger’s recent statement that the world is not broken into physical but informational parts.

## 1. Introduction

This work departs from the premise that physical objects are information-theoretic in origin, an idea that has recurrently appeared in human history, especially in physics, particularly in quantum physics research, for over a century. To further put this study in context with other studies and opinions, we would like to first consider the following views of great scientists, which depict a similar point of view.

For example, when asked about an underlying quantum world, N. Bohr, physicist and founder of quantum mechanics, answered: “There is no quantum world. There is only an abstract quantum physical description. It is wrong to think that the task of physics is to find out how Nature is. Physics concerns what we can say about Nature.” [1]. N. Bohr insisted that what we model results from observation, not the world itself.

Contemporary to N. Bohr, W. Heisenberg, physicist and founder of quantum mechanics, said: “The laws of nature which we formulate mathematically in quantum theory deal no longer with the particles themselves …,” but “…with our knowledge of the elementary particles.” [2]. For W. Heisenberg, the wave function codifies the knowledge about the system we study.

A few years later, in a brilliant paper in 1989, theoretical physicist J.A. Wheeler argued that information is the most fundamental thing, giving rise to the physical, a notion he summarized with the expression “it from bit.” Quoting him, “It from bit symbolizes the idea that every item of the physical world has at bottom—at a very deep bottom, in most instances—an immaterial source and explanation; that what we call reality arises in the last analysis from the posing of yes-no questions and the registering of equipment-evoked responses; in short, that all things physical are information-theoretic in origin and this is a participatory universe.” [3].

More recently, A. Zeilinger, quantum physicist and Nobel laureate in physics in 2022, has proposed an informational viewpoint to quantum mechanics in which the world is not broken up into physical but informational parts. According to him, a quantum system represents our knowledge of the world, not the world itself. Quoting him in a recent interview: “We can make measurements but cannot say anything about the essence of reality.” [4]. Therefore, we interpret his position as also in line with previous opinions.

On the other hand, according to neuroscience, we experience reality depending on interactions between ourselves, other beings, and multiple information sources. The nervous system’s different sensory modalities, smell, vision, audition, taste, and touch, generate evoked responses to the many questions we ask of the universe. One could say that sensory responses give rise to the phenomena we experience, with the brain being the natural organ capable of representing the surrounding reality. According to this view, physical observables—the “nature things”—may emerge after this preliminary process. We call this stage “pre-physics”, which refers to the preprocessing of the source data information performed, in principle, by our sensory systems and the brain.

With this initial work, we want to support and follow up on these relevant statements by renowned scientists about what we observe to be nothing but information by providing information-theoretic models for representing physical observables. In general, information-theoretic models are defined as theories designed to describe an entire behavior or specific situation with the idea that they will eventually be able to predict that behavior. We want to provide information-theoretic models to explain the “pre-physics” stage from which everything may emerge.

Toward this goal, we introduce a novel mathematical framework that can serve as the “informational” foundation for representing physical objects. The proposed approach is based on the field of information geometry, using the Fisher information metric as a particular Riemannian metric defined in the parameter space of a smooth statistical manifold of normal probability distributions. The informative geometry developed by C.R. Rao, extending results from P.C. Mahalanobis and R.A. Fisher, began in 1945 [5] but did not appear again until 1982 [6]. Since then, many works have highlighted its properties. The most relevant is the invariance against model transformations and changes in the model’s reference measurement, which makes it appropriate for addressing the formal properties of the observation process.

In developing the approach, we obtain exciting results about the building blocks of information and how they can be combined to represent the most likely reality. In this way, the laws of physics may come up afterward through a set of linear and nonlinear transformations of the parameters of an informative manifold. The approach is novel because it strives to lay down informational foundations for physics, especially quantum physics. The results are interesting because they connect fields such as information geometry, quantum physics, estimation, and probability theory. We certainly recognize that the project is very ambitious, but we have prospects for continuing work on future publications, making this article the first manuscript.

## 2. Mathematical Framework

The plan of this section, which, for didactic purposes, we divide into six subsections, is the following. In Section 2.1, we outline the modeling of a single source and the derivation of the Fisher information and the Riemannian manifold. Section 2.2 is devoted to analyzing the stationary states in the Riemannian manifold. In Section 2.3, we present the solutions of the stationary states in our formalism. In Section 2.4, we compute the probability density function, the estimator’s variance, and the Cramér–Rao lower bound for a single source. An extension of this approach to *m* independent sources is performed in Section 2.5 to compute the global probability density function at the ground-state level. In Section 2.6, we use Bayes’ theorem to obtain the posterior probability density function.

### 2.1. Single Source, the Fisher Information, and the Riemannian Manifold

We start our mathematical description by modeling a single source with a univariate normal probability distribution N(μ,σ) with σ=1 [7]. This is a well-known parametric statistical model whose parameter space may be identified with the real line, i.e., Θ=R. We can compute all the relevant quantities relevant to our purpose. Some steps may seem obvious to those used to information geometry.

For a single sample, the normal distribution, the log-normal (or lognormal) distribution, and the derivative (or gradient) of the lognormal distribution with respect to the parameter μ, respectively, are
(1)$$\begin{array}{ccc} f(x;\mu ) & = & \frac{1}{\sqrt{2\pi }}e^{-\frac{1}{2}{\left(x-\mu \right)}^{2}}\;, \end{array}$$
(2)$$\begin{array}{ccc} \ln f & = & -\frac{1}{2}\ln\left(2\pi \right)-\frac{1}{2}{\left(x-\mu \right)}^{2}\;, \end{array}$$
(3)$$\begin{array}{ccc} \frac{\partial \ln f}{\partial \mu } & = & x-\mu \;. \end{array}$$
Equation (3) is also called the score function. We can calculate Fisher’s information [8] as the variance of the score function:
(4)$$I\left(\mu \right)=E_{\mu }\left({\left(\frac{\partial \ln f}{\partial \mu }\right)}^{2}\right)=E_{\mu }\left({\left(x-\mu \right)}^{2}\right)=\int_{-\infty }^{\infty }{\left(x-\mu \right)}^{2}\frac{1}{\sqrt{2\pi }}e^{-\frac{1}{2}{\left(x-\mu \right)}^{2}}=1\;.$$
The Fisher information measures how much information is available about an informative parameter, in this case, the parameter μ. If the source generates *n* samples, $\left\{x^{1},\ldots ,x^{n}\right\}$, drawn independently from a univariate normal probability distribution (1), the likelihood distribution, the log-likelihood distribution, and the score function, respectively, are
(5)$$\begin{array}{ccc} f_{n}\left(x^{1},\ldots ,x^{n};\mu \right) & = & \prod_{i=1}^{n}f\left(x^{i};\mu \right)={\left(\frac{1}{2\pi }\right)}^{\frac{n}{2}}\exp\left(-\frac{1}{2}\sum_{i=1}^{n}{\left(x^{i}-\mu \right)}^{2}\right)\;, \end{array}$$
(6)$$\begin{array}{ccc} \ln f_{n} & = & -\frac{n}{2}\ln\left(2\pi \right)-\frac{1}{2}\sum_{i=1}^{n}{\left(x^{i}-\mu \right)}^{2}\;, \end{array}$$
(7)$$\begin{array}{ccc} \frac{\partial \ln f_{n}}{\partial \mu } & = & \sum_{i=1}^{n}\left(x^{i}-\mu \right)=n\left(\bar{x}-\mu \right)\;. \end{array}$$
Likewise, from Equation (7), we can calculate Fisher’s information for *n* samples as
(8)$$I_{n}\left(\mu \right)=E_{\mu }\left({\left(\frac{\partial \ln f_{n}}{\partial \mu }\right)}^{2}\right)=E_{\mu }\left(n^{2}{\left(\bar{x}-\mu \right)}^{2}\right)=n^{2}\int_{-\infty }^{\infty }{\left(\bar{x}-\mu \right)}^{2}{\left(\frac{n}{2\pi }\right)}^{\frac{1}{2}}e^{-\frac{1}{2}n{\left(\bar{x}-\mu \right)}^{2}}=n\;.$$
In other words,
(9)$$I_{n}\left(\mu \right)=nI\left(\mu \right)\;,$$
which shows the well-known additive property of the Fisher information for independent samples.

We can now introduce Riemannian manifolds [9]. These are smooth manifolds equipped with Riemannian metrics (smoothly varying choices of inner products on tangent spaces), which allow us to measure geometric quantities such as distances and angles.

The Riemannian metric from a single source with *n* samples derived from the Fisher information (8), is a scalar whose covariant component, contravariant component, and determinant, respectively, are
(10)$$\begin{array}{ccc} I_{n}\left(\mu \right)=g_{11}\left(\mu \right) & = & n\;, \end{array}$$
(11)$$\begin{array}{ccc} g^{11}\left(\mu \right) & = & \frac{1}{n}\;, \end{array}$$
(12)$$\begin{array}{ccc} \det(g(\mu )) & = & n\;. \end{array}$$

### 2.2. Stationary States in the Riemannian Manifold

To calculate the stationary states, we can invoke the time-independent non-relativistic Schrödinger’s equation [10] or the principle of minimum Fisher’s information [11]. The two approaches have been demonstrated to be equivalent [11]. The wave equation reads as follows:
(13)$$-k\nabla^{2}\psi \left(\mu \right)+U\left(\mu \right)\psi \left(\mu \right)=\lambda \psi \left(\mu \right)\;,$$
where U(μ) is the potential energy and k,λ are constants >0. The solution must also satisfy $\lim_{\mu \to -\infty }\psi \left(\mu \right)=\lim_{\mu \to +\infty }\psi \left(\mu \right)=0$ and $\int_{-\infty }^{\infty }\psi^{2}\left(\mu \right)d\mu =1$. For simplicity, we will write ψ instead of ψ(μ). We can use the modulus square of the score function (7) as the potential energy, except for a constant term:
(14a)$${∥\frac{\partial \ln f_{n}}{\partial \mu }∥}^{2}=n\left(\bar{x}-\mu \right)\;g^{11}\left(\mu \right)\;n\left(\bar{x}-\mu \right)\;,$$
(14b)$$=n\left(\bar{x}-\mu \right)\;\frac{1}{n}\;n\left(\bar{x}-\mu \right)\;,$$
(14c)$$=n{\left(\bar{x}-\mu \right)}^{2}\;.$$
Other approaches for the potential energy lead to similar expressions, such as considering $\ln f_{n}\left(x^{1},\ldots ,x^{n};\bar{x}\right)-\ln f_{n}\left(x^{1},\ldots ,x^{n};\mu \right)=\frac{n}{2}{\left(\bar{x}-\mu \right)}^{2}$, which is almost identical to Equation (14c). In this way, the potential energy can be written as $U\left(\mu \right)=nC{\left(\bar{x}-\mu \right)}^{2}$ where *C* is a constant >0. Equation (13) reads as
(15)$$-k\nabla^{2}\psi +nC{\left(\bar{x}-\mu \right)}^{2}\psi =\lambda \psi \;.$$
We compute the Laplacian in Equation (15) as
(16a)$$\nabla^{2}\psi =\frac{1}{\sqrt{\left|g(\mu )\right|}}\frac{\partial }{\partial \mu }\left(\sqrt{\left|g(\mu )\right|}\;g^{11}\left(\mu \right)\;\frac{\partial \psi }{\partial \mu }\right)\;,$$
(16b)$$=\frac{1}{\sqrt{n}}\frac{\partial }{\partial \mu }\left(\sqrt{n}\;\frac{1}{n}\;\frac{\partial \psi }{\partial \mu }\right)\;,$$
(16c)$$=\frac{1}{n}\frac{\partial^{2}\psi }{\partial \mu^{2}}=\frac{1}{n}\psi^{\prime \prime }\;.$$
Inserting Equation (16) into Equation (15), we obtain
(17)$$-\frac{k}{n}\psi^{\prime \prime }+nC{\left(\bar{x}-\mu \right)}^{2}\psi =\lambda \psi \;,$$
which is the Schrödinger’s equation of the quantum harmonic oscillator [12].

### 2.3. Solutions of the Quantum Harmonic Oscillator in the Riemannian Manifold

Some steps now may seem obvious for those used to quantum mechanics. Considering that ψ has the following form:
(18)$$\psi \left(\mu \right)=\gamma e^{\eta }\;,\;\;\;\;\mathrm{with}\;\gamma >0\;\mathrm{real}\;,\;\eta \;a\;\mathrm{function}\;\mathrm{of}\;\mu \;,$$
Equation (17) results in
(19a)$$-\frac{k}{n}\left(\gamma e^{\eta }{\left(\eta^{\prime }\right)}^{2}+\gamma e^{\eta }\eta^{\prime \prime }\right)+nC{\left(\bar{x}-\mu \right)}^{2}\gamma e^{\eta }=\lambda \gamma e^{\eta }\;,$$
(19b)$$-\frac{k}{n}\left({\left(\eta^{\prime }\right)}^{2}+\eta^{\prime \prime }\right)+nC{\left(\bar{x}-\mu \right)}^{2}=\lambda \;.$$
Assuming a solution for η(μ) with the form
(20)$$\eta \left(\mu \right)=-\xi n{\left(\bar{x}-\mu \right)}^{2}\;,\;\;\;\;\mathrm{with}\;\xi >0\;,$$
inserting this expression into Equation (19) gives
(21)$$-\frac{k}{n}\left(4\xi^{2}n^{2}{\left(\bar{x}-\mu \right)}^{2}-2\xi n\right)+nC{\left(\bar{x}-\mu \right)}^{2}=\lambda \;,$$
which implies that
(22)$$4k\xi^{2}n=nC\;,\;\;\;\;2k\xi =\lambda \;.$$
In other words, constants k,C,λ,ξ can not be chosen arbitrarily because they have to satisfy these equations. For example, we can choose $k=\frac{2}{n}$ and $C=\frac{1}{2n}$, which forces $\xi =\frac{1}{4}$, and $\lambda =\frac{1}{n}$. Therefore, we can write
(23)$$-\frac{2}{n^{2}}\psi^{\prime \prime }+\frac{1}{2}{\left(\bar{x}-\mu \right)}^{2}\psi =\frac{1}{n}\psi \;,$$
whose solution is given by
(24)$$\psi \left(\mu \right)=\gamma e^{-\frac{1}{4}n{\left(\bar{x}-\mu \right)}^{2}}\;.$$
With this configuration, we compute the normalization constant γ:
(25)$$1=\int_{-\infty }^{\infty }\psi^{2}\left(\mu \right)d\mu =\gamma^{2}\int_{-\infty }^{\infty }e^{-\frac{1}{2}n{\left(\bar{x}-\mu \right)}^{2}}d\mu =2\gamma^{2}\int_{0}^{\infty }e^{-\frac{1}{2}nt^{2}}dt\;,$$
where we used a first change of variable $\bar{x}-\mu =t$. Now, using a second change of variable $\frac{1}{2}nt^{2}=s$, $dt=\sqrt{\frac{2}{n}}\frac{1}{2}s^{-\frac{1}{2}}ds$, Equation (25) writes as
(26)$$1=2\gamma^{2}\int_{0}^{\infty }e^{-s}\sqrt{\frac{2}{n}}\frac{1}{2}s^{-\frac{1}{2}}ds=\gamma^{2}\sqrt{\frac{2}{n}}\int_{0}^{\infty }s^{-\frac{1}{2}}e^{-s}ds=\gamma^{2}\sqrt{\frac{2}{n}}\sqrt{\pi }\;.$$
Isolating γ from Equation (26), we obtain $\gamma ={\left(\frac{n}{2\pi }\right)}^{\frac{1}{4}}$. Therefore, Equation (24) reads as
(27)$$\psi \left(\mu \right)={\left(\frac{n}{2\pi }\right)}^{\frac{1}{4}}e^{-\frac{1}{4}n{\left(\bar{x}-\mu \right)}^{2}}=\psi_{0}\left(\mu \right)\;,$$
which is the ground-state solution of the quantum harmonic oscillator problem, the wave function for the ground state. The solutions of the quantum harmonic oscillator involve Hermite polynomials, which were introduced elsewhere [13,14]. In this way, we can prove, after some tedious but straightforward computations, that the wave function
(28)$$\psi_{1}\left(\mu \right)=\gamma_{1}\left(\bar{x}-\mu \right)e^{-\frac{1}{4}n{\left(\bar{x}-\mu \right)}^{2}}\;,\;\;\;\;\mathrm{with}\;\gamma_{1}>0\;,$$
is also a solution of
(29)$$-\frac{2}{n^{2}}\psi_{1}^{\prime \prime }+\frac{1}{2}{\left(\bar{x}-\mu \right)}^{2}\psi_{1}=\lambda_{1}\psi_{1}\;,$$
where $\lambda_{1}=\frac{3}{n}$ is the energy of the first excited state, and $\gamma_{1}={\left(\frac{n^{3}}{2\pi }\right)}^{\frac{1}{4}}$ is the normalization constant. With this representation, the λ’s (energy levels) are given by
(30)$$\lambda_{\nu }=\frac{2}{n}\left(\nu +\frac{1}{2}\right)=E_{\nu }\;,\;\;\;\;\;\mathrm{with}\;\nu =0,1,\ldots \;.$$
Looking closely at Equation (30), we appreciate that the energy levels depend on two numbers, ν and *n*. The ground state at ν=0 has a finite energy $E_{0}=\frac{1}{n}$, and can get arbitrarily close to zero by massive sampling.

### 2.4. Probability Density Function, Estimator’s Variance, and Cramér–Rao Lower Bound

Assuming that the square modulus of the wave function can be interpreted as the probability density function
(31)$${∥\psi \left(\mu \right)∥}^{2}=\psi^{*}\left(\mu \right)\psi \left(\mu \right)=\rho \left(\mu \right)\;,$$
we can compute the performance of the estimations of μ. For instance, we can calculate the variance of the estimator $\bar{x}$ at the ground state (27):
(32)$$E_{\mu ,\rho_{0}\left(\mu \right)}\left({\left(\bar{x}-\mu \right)}^{2}\right)=\frac{1}{n}\;.$$
Likewise, we can compute the variance of the estimator $\bar{x}$ at the first excited state (28):
(33)$$E_{\mu ,\rho_{1}\left(\mu \right)}\left({\left(\bar{x}-\mu \right)}^{2}\right)=\frac{3}{n}\;.$$
The estimator’s variance equals the quantum harmonic oscillator’s energy levels, i.e., the estimator’s variance is definitively quantized. Interestingly, the estimator’s variance at the ground state (32) equals the Cramér–Rao lower bound (CRLB) [5,15] on the variance of the estimator $\bar{x}$:
(34)$$E_{\mu }\left({\left(\bar{x}-\mu \right)}^{2}\right)\geq \frac{1}{I_{n}^{Q}\left(\mu \right)}=\frac{1}{I_{n}\left(\mu \right)}=\frac{1}{n}\;,$$
where $I_{n}^{Q}\left(\mu \right)$ is the quantum Fisher information for *n* samples, and $I_{n}\left(\mu \right)$ is the classical Fisher information for *n* samples, as computed in Equation (8). The quantum Fisher information coincides with the classical Fisher information when in a polar representation of the wave function $\psi \left(\mu \right)=\sqrt{\rho (\mu )}\exp\{i\alpha (\mu )\}$, α(μ)=0 [16], which is the case of study.
To summarize the findings, we reach the minimum variance at the minimum energy level. It is worth highlighting that the above equations entail a novel relation between energy and information. The energy is minimal at the ground-state level and increases if we do not have enough information or lose information. Notably, the loss of information is quantized. This phenomenon deserves to be further explored in forthcoming communications.

### 2.5. m Independent Sources and Global Probability Density Function

With *m* independent sources, each generating *n* samples, a finite set of *m* quantum harmonic oscillators may represent reality. Presuming independence of the sources of information, the “global” wave function (also called the collective wave function) can factor as the product of single wave functions. We can write the global wave function as
(35)$$\psi \left(\mu \right)=\psi \left(\mu^{1},\ldots ,\mu^{m}\right)=\psi \left(\mu^{1}\right)\ldots \psi \left(\mu^{m}\right)\;.$$
When the sources do not evolve independently, the collective wave function for an *m*-system cannot be written as the product of individual wave functions; in this case, we speak of entangled states.

Equation (35) constitutes a many-body system, and we may refer to the vector μ as the μ field. For example, in the case of modeling two independent sources, the global wave function at the ground state will be the product of single wave functions, each of them at the ground state:
(36a)$$\psi_{0}\left(\mu \right)=\psi_{0}\left(\mu^{1}\right)\psi_{0}\left(\mu^{2}\right)\;,$$
(36b)$$={\left(\frac{n}{2\pi }\right)}^{\frac{1}{4}}\exp\{-\frac{1}{4}n{\left({\bar{x}}^{1}-\mu^{1}\right)}^{2}\}{\left(\frac{n}{2\pi }\right)}^{\frac{1}{4}}\exp\{-\frac{1}{4}n{\left({\bar{x}}^{2}-\mu^{2}\right)}^{2}\}\;,$$
(36c)$$={\left(\frac{n}{2\pi }\right)}^{\frac{1}{4}}{\left(\frac{n}{2\pi }\right)}^{\frac{1}{4}}\exp\{-\frac{1}{4}\left(n{\left({\bar{x}}^{1}-\mu^{1}\right)}^{2}+n{\left({\bar{x}}^{2}-\mu^{2}\right)}^{2}\right)\}\;,$$
(36d)$$={\left(\frac{n}{2\pi }\right)}^{\frac{1}{2}}\exp\{-\frac{1}{4}n\left(\left({\bar{x}}^{1}-\mu^{1},{\bar{x}}^{2}-\mu^{2}\right)\left(\begin{array}{c} {\bar{x}}^{1}-\mu^{1}\\ {\bar{x}}^{2}-\mu^{2} \end{array}\right)\right)\}\;.$$
We can generalize Equation (36) for having *m* independent sources. The global wave function writes as
(37)$$\psi_{0}\left(\mu \right)={\left(\frac{n}{2\pi }\right)}^{\frac{m}{4}}\exp\{-\frac{1}{4}n{\left(\bar{x}-\mu \right)}^{T}\left(\bar{x}-\mu \right)\}\;.$$
Using Equation (31), the global probability density function is
(38)$$\rho_{0}\left(\mu \right)={\left(\frac{n}{2\pi }\right)}^{\frac{m}{2}}\exp\{-\frac{1}{2}n{\left(\bar{x}-\mu \right)}^{T}\left(\bar{x}-\mu \right)\}\;,$$
which “essentially” results from Schrödinger’s equation (or the principle of minimum Fisher’s information) on the Riemannian manifold assuming independence of sources.

### 2.6. Bayesian Framework and Posterior Probability Density Function

In parallel, we can compute, using Bayes’ theorem [17], the posterior probability density function from the *m* sources of information for all data values, taking the Riemannian volume of the metric as a prior probability distribution. As proven elsewhere, the Riemannian volume can be considered an objective choice for a prior probability distribution [18]. In honor of the author, this measure is called the Jeffreys’ prior probability distribution on informative parameters.

To start with, we can model *m* independent sources with a multivariate normal probability distribution $N_{m}\left(\mu ,\Sigma \right)$ with $\Sigma =I_{m}$. Some developments may now be trivial for those used in probability theory. If every source generates *n* samples that are drawn independently from the multivariate normal probability distribution, the likelihood probability distribution is
(39a)$$f_{m,n}\left(x^{1},\ldots ,x^{n};\mu \right)=\prod_{i=1}^{n}\frac{1}{{\left(2\pi \right)}^{\frac{m}{2}}}{\left(\det\left(I_{m}\right)\right)}^{-\frac{1}{2}}\exp\{-\frac{1}{2}\left({\left(x^{i}-\mu \right)}^{T}I_{m}^{-\frac{1}{2}}\left(x^{i}-\mu \right)\right)\}\;,$$
(39b)$$={\left(2\pi \right)}^{-\frac{mn}{2}}\exp\{-\frac{1}{2}\sum_{i=1}^{n}{\left(x^{i}-\mu \right)}^{T}\left(x^{i}-\mu \right)\}\;,$$
(39c)$$={\left(2\pi \right)}^{-\frac{mn}{2}}\exp\{-\frac{1}{2}\mathrm{Tr}\left(\sum_{i=1}^{n}\left(x^{i}-\mu \right){\left(x^{i}-\mu \right)}^{T}\right)\}\;.$$
The summation term within the trace operator can be decomposed into two terms:
(40a)$$\sum_{i=1}^{n}\left(x^{i}-\mu \right){\left(x^{i}-\mu \right)}^{T}=\sum_{i=1}^{n}\left(x^{i}-\bar{x}\right){\left(x^{i}-\bar{x}\right)}^{T}+n\left(\bar{x}-\mu \right){\left(\bar{x}-\mu \right)}^{T}\;,$$
(40b)$$=nS+n\left(\bar{x}-\mu \right){\left(\bar{x}-\mu \right)}^{T}\;,$$
where S is the covariance matrix, a symmetric constant matrix. Inserting Equation (40) into Equation (39) gives
(41)$$f_{m,n}\left(x^{1},\ldots ,x^{n};\mu \right)={\left(2\pi \right)}^{-\frac{mn}{2}}\exp\{-\frac{1}{2}n\mathrm{Tr}\left(S\right)\}\exp\{-\frac{1}{2}n{\left(\bar{x}-\mu \right)}^{T}\left(\bar{x}-\mu \right)\}\;.$$
To compute the posterior probability distribution, we choose a prior probability distribution proportional to the Riemannian volume:
(42)$$\sqrt{\det g(\mu )}=n^{\frac{m}{2}}\;.$$
Thus, $p\left(\mu \right)\propto n^{\frac{m}{2}}$. The posterior probability distribution will be proportional to the likelihood probability distribution (41) multiplied by the prior probability distribution (42):
(43a)$$f_{m,n}\left(\mu ;x^{1},\ldots ,x^{n}\right)\propto f_{m,n}\left(x^{1},\ldots ,x^{n};\mu \right)p\left(\mu \right)\;,$$
(43b)$$\propto {\left(2\pi \right)}^{-\frac{mn}{2}}\exp\{-\frac{1}{2}n\mathrm{Tr}\left(S\right)\}\exp\{-\frac{1}{2}n{\left(\bar{x}-\mu \right)}^{T}\left(\bar{x}-\mu \right)\}n^{\frac{m}{2}}\;,$$
(43c)$$\propto \underset{\mathrm{constant}\;\mathrm{term}}{\underset{︸}{{\left(2\pi \right)}^{-\frac{mn}{2}}\exp\{-\frac{1}{2}n\mathrm{Tr}\left(S\right)\}n^{\frac{m}{2}}}}\underset{\mu \text{-dependent}\;\mathrm{term}}{\underset{︸}{\exp\{-\frac{1}{2}n{\left(\bar{x}-\mu \right)}^{T}\left(\bar{x}-\mu \right)\}}}\;,$$
(43d)$$\propto \exp\{-\frac{1}{2}n{\left(\bar{x}-\mu \right)}^{T}\left(\bar{x}-\mu \right)\}\;.$$
We need to normalize Equation (43) to be a probability density function:
(44a)$$1=\int_{R^{m}}\exp\{-\frac{1}{2}n{\left(\bar{x}-\mu \right)}^{T}\left(\bar{x}-\mu \right)\}\;d\mu^{1}\ldots d\mu^{m}\;,$$
(44b)$$=\int_{R^{m}}\prod_{i=1}^{m}\exp\{-\frac{1}{2}n{\left({\bar{x}}^{i}-\mu^{i}\right)}^{2}\}\;d\mu^{1}\ldots d\mu^{m}\;,$$
(44c)$$={\left\{\int_{-\infty }^{\infty }\exp\{-\frac{1}{2}n{\left({\bar{x}}^{i}-\mu^{i}\right)}^{2}\}\;d\mu^{i}\right\}}^{m}\;,$$
(44d)$$={\left\{2\int_{0}^{\infty }\exp\{-\frac{1}{2}n{\left({\bar{x}}^{i}-\mu^{i}\right)}^{2}\}\;d\mu^{i}\right\}}^{m}\;.$$
Using a change of variable $t=\frac{1}{2}n{\left(\mu^{i}-{\bar{x}}^{i}\right)}^{2}$, ${\bar{x}}^{i}+\sqrt{\frac{2}{n}t}=\mu^{i}$, $d\mu^{i}=\sqrt{\frac{2}{n}}\frac{1}{2}t^{-\frac{1}{2}}dt$, Equation (44) reads as
(45)$$1={\left\{2\int_{0}^{\infty }\exp\{-t\}\frac{1}{\sqrt{2n}}t^{-\frac{1}{2}}\;dt\right\}}^{m}={\left(\frac{\sqrt{2}}{\sqrt{n}}\Gamma \left(\frac{1}{2}\right)\right)}^{m}={\left(\frac{2\pi }{n}\right)}^{\frac{m}{2}}\;.$$
Thus, the normalization constant is ${\left(\frac{n}{2\pi }\right)}^{\frac{m}{2}}$, and the posterior probability density function (43) is
(46)$$f_{m,n}\left(\mu ;x^{1},\ldots ,x^{n}\right)={\left(\frac{n}{2\pi }\right)}^{\frac{m}{2}}\exp\{-\frac{1}{2}n{\left(\bar{x}-\mu \right)}^{T}\left(\bar{x}-\mu \right)\}\;,$$
which is the global probability density function of the global wave function at the ground state (38). Precisely, the global probability density function of *m* quantum harmonic oscillators at the ground state corresponds to the posterior probability density function calculated using Bayes’ theorem from *m* sources of information for all data values. Interestingly, the opposite is also true, as the prior is constant (42). In other words, we can also say that the likelihood probability distribution from *m* sources of information for all data values equals the posterior probability density function calculated using Bayes’ theorem from *m* quantum harmonic oscillators at the ground state. This unexpected and exciting result reveals a plausible relationship between energy and Bayes’ theorem.

## 3. Discussion

Over the past century, renowned scientists have proposed that what we observe is nothing but information. Among many scientists’ opinions, we find that what we actually model is the result of observation, not the world itself (N. Bohr); the laws of nature do not deal with the particles themselves but with our knowledge of elementary particles (W. Heisenberg); all things physical are information-theoretic in origin (J.A. Wheeler); and the world is not broken up into physical but informational parts (A. Zeilinger). From a pure neuroscientific perspective, we can fairly say that we experience reality depending on the central and peripheral nervous system interactions with multiple information sources. In this way, the physical observables—the “nature things”—may emerge after pre-processing the source data information performed by our sensory systems and the brain, a stage we call “pre-physics”, with the laws of physics appearing afterward.

With this initial work, we wanted to provide information-theoretic models to explain this generative process. We developed a novel mathematical framework for representing physical objects. Our approach is based on information geometry and uses Fisher’s information metric as a Riemannian metric, defined on a smooth statistical manifold of normal probability distributions. We assumed that a single information source generating *n* samples can be modeled as a univariate normal probability distribution N(μ,σ), setting σ=1 for simplicity. With this modeling, the Fisher information equals *n*, the number of samples. We also modeled *m* information sources with a multivariate normal probability distribution. It is an approximation for apparent reasons, and we may explore other probability distributions.

Investigating the stationary states invoking Schrödinger’s equation or the principle of minimum Fisher’s information, we discovered that information could be represented and distributed over a set of quantum harmonic oscillators, one for each independent source, whose coordinate is the μ parameter of the statistical manifold to estimate, taking into consideration the assumptions of our modeling. We observed that the estimator’s variance equals the energy levels of the quantum harmonic oscillator, proving that the estimator’s variance is definitively quantized. The minimum estimator’s variance occurs at the minimum energy level of the oscillator, which equals $\frac{1}{n}$, where *n* is the number of samples. Also, we demonstrated that quantum harmonic oscillators reach the Cramér–Rao lower bound on the estimator’s variance at the lowest energy level, which indicates a relationship between energy and information that deserves to be studied in detail in future work.

We then showed that the global probability density function of the collective mode of a set of *m* quantum harmonic oscillators at the lowest energy level, calculated as the square modulus of the global wave function at the ground state, equals the posterior probability distribution calculated using Bayes’ theorem from the *m* sources of information for all data values, taking as a prior the Riemannian volume of the informative metric. Interestingly, the opposite is also true, as the prior is constant. These results suggest that the *m* sources of information could be broken into *m* little elements, in particular, *m* quantum harmonic oscillators, with the square modulus of the collective mode at the lowest energy representing the most likely reality, which supports A. Zeilinger’s statement that the world is not broken into physical but informational parts. In addition, these findings reveal a very interesting connection between energy and Bayes’ theorem to be explored in the future.

Interestingly, apart from the theoretical results we reported to be further explored, such as the relation between energy and information and the connection between energy and Bayes’ theorem, respectively, this mathematical framework offers other multiple alternatives that we are currently exploring. For example, the informational representation on statistical manifolds with σ unknown and the representation of multiple “dependent” sources. Moreover, in forthcoming studies, we will strive to generalize this approach by exploring other statistical manifolds and to depict how physical observables such as space and time may emerge from linear and nonlinear transformations of a set of parameters of a specific statistical manifold. This way, the laws of physics, including time’s arrow, will appear afterward.

Indeed, we can also speculate about the possibility of physical entities being represented this way, for example, in our nervous system. This complex system, consisting of several components, including water molecules, cell types, and conduction systems, could harbor this representation of information. However, we will not discuss these topics in this work. We will leave it now for internal conversations in which we extensively discuss how we may sample and perceive reality, which is possibly the true motivation for initiating and developing this research project.

## Data Availability

No new data were created.

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
