# Peer review of "Information-Theoretic Models for Physical Observables"

_entropy, 2023, doi:10.3390/e25101448_

Round 1
Reviewer 1 Report
I do not recommend the publication of the paper. The authors presented some standard and quite straightforward calculations concerning the normal distribution accompanied by some claims concerning the novelty of their approach. Perhaps, the authors could improve and strengthen their message by writing explicitly: 1) what is novel in their approach 2) why their results are interesting.
The connections to the views of Bohr, Heisenberg, and Wheeler are completely unclear and hardly substantiated in the paper.
The authors mention several times that their approach is based on the "information geometry". Indeed they use such terms as "Riemanian metric" and "Riemann volume" but without any explanation why these notions are appropriate in the context. The notion of "information geometry" is well established and it is the topic of many monographs, e.g. those authored by Shun-ichi Amari or Nihat Ay et. al. Maybe a reference to at least one of them would be beneficial for potential readers. There is no such reference in the paper.
Author Response
Please see PDF attached.

Reviewer 2 Report
The paper is certainly interesting, but obscure.
The authors need many more detailed and more detailed explanatios.
Also, be careful in defining ALL symols that they use.
They cite mostly fundamental, old refereces and not more recent papers.
For example, Frieden book and NOT his papers, that are much more concise.
No one will loose hours looking into the book for what the authors mention.
As it is, the paper is very difficult to read.
The authors should work on the didactis.
Author Response
Please see PDF attached.

Reviewer 3 Report
My comments are in pdf format.

Author Response
Please see PDF attached.

Round 2
Reviewer 1 Report
I still do not understand what is the main message of the paper and what is novel in all calculations (correct) presented. In particular, what are the reasons for invoking Riemannian structures (trivial in this case).
Since the authors claim that the paper can be a basis for further interesting research and that the problems raised by other referees were of rather technical character (and those are addressed rather properly by the authors) I do not object to accepting the paper, hoping that it will really be a starting point to something more profound.
Reviewer 2 Report
This is an important an instructive paper.
Please, accept.